# Protocol for a scoping review to understand what is known about how GPs make decisions with, for and on behalf of patients who lack capacity

Simon Jack Ogden [1], Richard Huxtable,[2] Jonathan Ives[1]

¹Centre for Ethics in Medicine, University of Bristol, Bristol, UK
²School of Social and Community Medicine, University of Bristol, Bristol, UK

**Correspondence to**
Dr Simon Jack Ogden;
jack.ogden@bristol.ac.uk

## ABSTRACT

**Introduction** General Practitioners (GPs) and allied healthcare professionals working in primary care are regularly required to make decisions with, for and on behalf of patients who lack capacity. In England and Wales, these decisions are made for incapacitated adult patients under the Mental Capacity Act 2005, which primarily requires that decisions are made in the patient's 'best interests'. Regarding children, decisions are also made in their best interests but are done so under the Children Act 1989, which places paramount importance on the welfare of the child. Decisions for children are usually made by parents, but a GP may become involved if he or she feels a parent is not acting in the best interests of the child. Internationally, including elsewhere in the UK, different approaches are taken. We hypothesise that, despite the legislation and professional guidelines, there are many different approaches taken by GPs and allied healthcare professionals in England and Wales when making these complex decisions with, for and on behalf of patients who lack capacity. To better understand what is known about how these decisions are made, we plan to undertake a scoping review and directed content analysis of the literature. While the majority of decisions made in primary care are made by GPs, for completeness, this review will include all allied healthcare professionals working in primary care.

**Methods and analysis** To ensure a wide breadth of literature is captured, a scoping review will be undertaken as described by Arksey and O'Malley (2005). A five-stage approach will be taken when conducting this review: (1) identifying the research question; (2) identifying relevant papers; (3) study selection; (4) data extraction and (5) summarising and synthesis. The final stage will include a directed content analysis of the data to help establish the cross-cutting themes.

**Ethics and dissemination** The scoping review will be disseminated through conferences and peer-reviewed publications. This scoping review is the first (mapping) phase in a proposed larger study to explore how GPs make decisions with, for and on behalf of those who lack capacity. Qualitative research with GPs, patients and their families will follow, before all the results are synthesised using an 'empirical bioethics' methodology.

## Strengths and limitations of this study

► This scoping review will map the current understanding of how decisions are or should be made in primary care with, for and on behalf of patients who lack capacity.
► Arksey and O'Malley's scoping review methodology will be used and it will follow the Preferred Reporting Items for Systematic Reviews and Meta-Analyses Extension for Scoping Reviews checklist.
► An international search will allow comparison of approaches between different legal jurisdictions.
► The data will be critically synthesised with reference to ethical and legal principles to identify cross-cutting themes and to ascertain where further research is needed on this increasingly important topic.

has been said that the NHS is a victim of its own success; with novel and improved treatments, patients live longer, although with higher rates of comorbidity and medical complexity, often with reduced capacity to make their own medical decisions.[4] As a result, GPs are seeing an increasing number of patients who are unable to make their own decisions about their medical treatment.[5] Some common conditions that often result in decisions being made on behalf of the patient include dementia, learning disabilities, acquired traumatic brain injuries and mental health conditions. In addition to medical decisions, GPs are also often involved in decisions concerning the location of a patient's care, be that at home, in hospital or in a care home.

In England and Wales, the Mental Capacity Act (MCA) 2005[6] provides a legal framework for making decisions for patients who lack capacity, that is, are unable to do so themselves. The assessment of capacity under the MCA can be made by any professional involved in the patient's care; in primary care, this will often be a GP, although in

## INTRODUCTION

General Practitioners (GPs) in the UK National Health Service (NHS) provide over 300 million consultations per year[1] and the workload continues to rise year-on-year.[2 3] It

care homes, for example, this may be an allied healthcare professional. The MCA sets out a two-stage test of capacity. First, the diagnostic test asks whether the person has an impairment of their mind or brain, whether as a result of an illness, or external factors such as alcohol or drug use. Second, the functional test asks whether the impairment means that the person is unable to make a specific decision when they need to.

The MCA primarily states that if a patient lacks the capacity to make a particular decision, then it can be made for him or her in his or her 'best interests' and must be done in a way that is least restrictive of their rights. Although the act does not provide a formula for best interests decision-making, the act does disclose some of the factors that must be considered when making these best interest decisions, including the patient's past and present wishes and feelings, the beliefs and values that would likely influence their decision if they had capacity and other factors that they would be likely to consider if they were able to do so. There is also a duty to consult with those close to the patient, such as his or her family, when making best interests decisions.

The MCA also empowers patients to make provision for a future in which they may lack the capacity to make decisions. Two options are available. First, the MCA provides for proxy decision-making by introducing 'lasting power of attorneys', who can make decisions on behalf of patients when they lose capacity. However, these proxies must ultimately still conform to the rules set out in the MCA with regards to acting in the patient's best interests. Second, the MCA places advance decision-making on a statutory footing, by introducing the 'advance decision to refuse treatment', which may refuse (but not consent to or demand) specified treatments that might otherwise be provided after the patient has lost capacity.

Anglo-Welsh law regarding the capacity to consent and refuse treatment in children is slightly more complicated. The treatment of children is generally governed by the Children Act 1989 (CA), which requires that the paramount consideration is the welfare, or 'best interests', of the child. The Family Law Reform Act 1969 states that those aged 16 and 17 years of age have the same legal ability to consent as an adult; however, this does not mean that they have a right to refuse treatment Re W (A Minor)[7] and indeed someone with parental responsibility (or a court) may consent on their behalf. In children younger than 16, the law surrounding their capacity to consent is set out in Gillick v West Norfolk and Wisbech AHA.[8] The term 'Gillick competence' refers to a child under 16 who 'reaches a sufficient understanding and intelligence to be capable of making up his own mind on the matter requiring decision'.[9] As with older children, even a Gillick competent minor cannot expect his refusal of treatment to operate as a veto. As such, where a child refuses to consent, or lacks the competence to consent, decisions may be made by others, in the 'best interests' of the child.

Internationally, different legal jurisdictions provide different approaches to making decisions with, for and on behalf of patients who lack capacity to make these decisions themselves. For example, in Scotland, patients are treated under the Adults with Incapacity (Scotland) Act 2000 where a doctor is required to complete a 'section 47' certificate if a patient is deemed to lack capacity and cannot use force unless immediately necessary. Conspicuous is the absence of the term 'best interests' in the Adults with Incapacity (Scotland) Act, instead it chooses to set out five principles that must be applied when making decisions. Any action or decision must benefit the person, it must be the least-restrictive option, take account of the wishes of the person, include a consultation with a relevant other and encourage the person to use existing skills and develop new skills. In Northern Ireland, patients are treated under the Mental Capacity (Northern Ireland) Act 2016 which covers the treatment of all patients who lack capacity, including those in whom incapacity arises as a result of a mental health condition. This is different to England and Wales, where there is separate legislation— the Mental Health Act 2007—governing the treatment of those with mental disorders. It is, however, similar in that it uses the 'best interests' test to guide how these decisions should be made. In the USA, decisions may be made by a nominated surrogate decision-maker, such as a family member.[10]

The United Nations Convention on the Rights of Persons with Disabilities is also of importance given that it was ratified by the UK in 2009 and is hailed by some as advocating for a supported decision-making approach[11] rather than a substituted or surrogate decision-making approach of which best interests decisions are an example. The UN Committee on the Rights of Persons with Disabilities stated that 'perceived or actual deficits in mental capacity must not be used as justification for denying legal capacity'.[12] It is explicitly stated by this UN committee that best interests decisions are substituted decisions and should be replaced by supported decisions.

Returning to England and Wales, the law on best interest decisions has been further criticised as being vague, making its application open to divergent interpretations.[13] Neither act—the MCA or the CA—elaborates a particular account of what is in a patient's best interests; rather, a non-exhaustive list of different factors is enumerated.[14 15] Different stakeholders and proxy decision-makers—such as clinicians, those with lasting powers of attorney, or parents—might point to different factors and reach different decisions, even in seemingly similar cases. We hypothesise that, despite the legislation and professional guidelines,[16–18] there are many different approaches taken by GPs and allied healthcare professionals when making these complex decisions depending on their interpretation of the facts, their own moral judgement and their own understanding of the patient in front of them, with some (but not all) of whom they might have a longstanding relationship.

## STUDY RATIONALE

With the increasing incidence of patients[5] who lack capacity to make decisions themselves comes an increasing need to understand how these decisions are made by GPs and allied healthcare professionals and how this process might be improved.

GPs have a unique role in the healthcare of patients and are often involved in the care of both the patient and their family, which can sometimes be for many years, from childhood to adulthood. GPs are also responsible for coordinating a patient's secondary care and managing most of their chronic diseases in the community setting.[19] While continuity of care has suffered in recent years, there is still a continuity in the relationship between GP and patient, which makes GPs well-placed to be involved in many of the complex medical decisions for patients who lack capacity.[20] The decisions being made could range from what treatment the patient should receive at the end of their life to whether a patient should be admitted to hospital or a care home.[21] Given these are often important decisions that have a significant impact both for the patient and their family, it is important to understand how and why these decisions are made. The scoping review aims to find out what is understood about how decisions are or should be made in primary care with, for and on behalf of patients who lack capacity.

## STUDY OBJECTIVES

The objective of this scoping review is to identify and map current research on how decisions are made with, for and on behalf of patients who lack capacity in primary care. The main concepts, approaches and theories will be established and synthesised with reference to current ethical principles and the various legal positions internationally. The overall objective is to identify knowledge gaps in the literature and areas where further research is needed.

## METHODS AND ANALYSIS
### Protocol design

A scoping review will be performed as described by Arksey and O'Malley[22] and set out in the Preferred Reporting Items for Systematic Reviews and Meta-Analyses guidelines by Tricco *et al*.[23] A scoping review methodology has been chosen as it will allow us to identify what is known about the key characteristics and factors that are important to decisions made with, for and on behalf of patients who lack capacity. Following initial searches, it was clear that there is limited research currently published in this area, making a scoping review an appropriate approach. Scoping reviews map the topic as a whole and highlight where gaps in knowledge exist, thus helping guide further research.[24] The aim is to understand the breadth and depth of existing research knowledge.[25]

According to Arksey and O'Malley, there are six stages to undertaking a scoping review: identify the research question; identify relevant studies; study selection; data collection; summarising, and finally, expert consultation. The first five of these stages will be applied in this scoping review. The final consultation stage will be omitted as this will form part of the next phase of the research. The time frame for completing the scoping review is 6 months.

### Stage 1: identify research question

The research question was formulated in consultation with the research team and with consideration given to the further planned 'empirical bioethics' research. 'Empirical bioethics' combines theoretical and empirical research and may take various forms, but here will involve mapping what is currently known through literature reviews (this phase),[26] empirical research to deepen our understanding, then shaping work, to recommend ways forward.[27] The question posed for the current mapping phase was intentionally kept broad in its wording so as to capture all the published research that speaks to this question, including both empirical and theoretical research. The main research question is:

► What is understood about how decisions are or should be made in primary care with, for and on behalf of patients who lack capacity?

Subordinate research questions include:

1. What guidelines, approaches and processes are used in making decisions in primary care with, for and on behalf of patients who lack capacity?
2. Who participates in, or is consulted about, decisions in primary care with, for and on behalf of patients who lack capacity?
3. Which factors are considered by the participants when making decisions in primary care with, for and on behalf of patients who lack capacity?
4. What does the theoretical literature reveal about how decisions are or should be made in primary care with, for and on behalf of patients who lack capacity?

### Stage 2: identify relevant studies—search strategy

A limited initial search was carried out using MEDLINE (PubMed) and Web of Science to discover frequently used search terms, abbreviations, keywords and MeSH terms. This stage is important so that terms that might be used in different countries and disciplines are discovered and included in the final search string to capture all relevant research. A subject librarian also advised on the proposed search terms and approach.

Pilot searches were performed to test various search strings using different combinations of keywords in addition to wildcards (*) and Boolean modifiers to produce the final search string. The aim of the pilot searches was to capture as much of the relevant research as possible.

The article title and abstracts of papers will be searched in MEDLINE, Embase, PsycINFO, The Cochrane Library, Web of Science and JSTOR using the terms in the final search string:

[gp OR general pract* OR primary care] AND
[capacity or compet* or incapacit*] AND

[best interest* or decision making or decision]

## Stage 3: study selection

The title and abstracts of all the papers returned from the search will be transferred to Endnote software where they will be screened by the lead reviewer (SJO) according to the inclusion and exclusion criteria below. Papers will be classified into 'include', 'exclude' and 'unsure' groups. Duplicated papers will be removed.

A second reviewer (RH) will screen 10% of the included and excluded papers to aid reliability. The second reviewer will also review all those papers in the 'unsure' group. Any disagreements that cannot be resolved by discussion will be resolved by bringing in a third reviewer (JI). The team will meet regularly to discuss results and will seek to reach consensus on the papers to include and exclude. Where consensus is not reached, a majority decision will be carried (ie, include or exclude).

Once the list of the included papers is agreed, the first reviewer will screen the references of the included papers to check for further papers that might fit the inclusion criteria. Any additional papers found will repeat the process above, as needed.

All papers that are found to fit the inclusion and exclusion criteria will be included in the scoping review.

### Inclusion

► Peer-reviewed papers, books, book chapters and national guidelines.
► All types of empirical research including qualitative, quantitative and mixed methods.
► Theoretical studies.
► Commentary or opinion pieces in peer-reviewed journals.
► Any paper that addresses the question of how medical decisions are made for patients who lack capacity and what factors are considered important. It is not necessary for this to be the paper's primary aim.
► Papers situated in the primary healthcare setting.
► Papers that consider decisions about healthcare treatment/non-treatment including decision about the location of a patients care.
► Papers where decisions are made following advance care planning when patients were considered to have capacity.
► Any papers published since 1960 and in English.

### Exclusion

► Grey literature including journalism.
► Commentary or opinion pieces not in peer-reviewed journals.
► Papers that do not refer to lack of capacity or, in the case of a child, competence when referring to any medical decision-making.
► Papers not situated in the primary healthcare setting.
► Papers that do not consider decisions about healthcare treatment/non-treatment, for example, that

concern research and/or social care exclusively. Papers not in English.

## Stage 4: data extraction and analysis

A directed content analysis will be undertaken in order to identify and describe key themes present in the papers. A directed content analysis begins with a pre-established coding framework, but then allows new codes to be used, or existing codes to be further specified, as the analysis progresses.[28] Directed content analysis will be performed so that the analysis can be directly guided by the search for ethical material—with codes relating to ethical content around best interests decisions. These codes will be developed by the first reviewer with regular discussion with the second and third reviewers. Where there are disagreements over coding the majority decision will be carried. While synthesis of the data is not always performed in scoping reviews, it was felt it would add context to the research included in the review.

The first stage will be data extraction (or 'charting the results')—an iterative process that involves extracting predefined data into a charting sheet, which can then be used to explore themes. The draft charting table comprises key details about the papers, but also includes the initial coding framework that will be applied to papers—focussing on four areas of results (see point 7 below). A description of what those papers say about each of those areas will be entered into the data extraction sheet, and then further subcodes will be developed, as appropriate, to further distinguish between, and record, different kinds of content.

1. Articles details.
2. Country of origin.
3. Broad category (ie, patient situation/condition).
4. Brief summary.
5. Aims of study.
6. Methods/type of study.
7. Results:
   a. Approaches and processes used in making decisions.
   b. Participants involved in making decisions.
   c. Factors considered in making decisions.
   d. Theoretical argument(s) about how to make decisions.
8. Conclusion.
9. Reflections from reviewer.

The first reviewer (SJO) will extract data from all included empirical and theoretical research into the charting table. Data from quantitative research will be described and charted. To improve reliability, 10%–15% of papers will also have data extracted independently by a second (RH) and third (JI) reviewer.

The first reviewer will extract data from all included empirical and theoretical research into the charting table. Data from quantitative research will be described and charted. To improve reliability, 10%–15% of papers will also have data extracted independently by a second and third reviewer.

The extracted data and expanded coding will be discussed among the group early in the data extraction stage to discuss disagreements and agree on how the coding framework should be developed and applied.

### Stage 5: summarising and synthesis

The second phase in the analysis will be to systematically explore the codes developed (and data extracted) in stage 4 to develop larger descriptive themes that summarise what the evidence from the papers show.

Any quantitative evidence will be simply described and reported narratively.

Finally, consideration will be given to what gaps in the literature exist and how these might inform further research on the topic.

### PATIENT AND PUBLIC INVOLVEMENT

No patients involved.

### ETHICS AND DISSEMINATION

One paper conveying the results of the review will be prepared for publication in a peer-reviewed journal. The authors will also seek to present the findings at appropriate bioethics or primary care conferences in the UK or virtual conferences internationally. The results from this review will help guide the next phase of the research involving a qualitative 'empirical bioethics' study, looking further into how GPs do and should make these decisions.

No ethics approval is required for this scoping review.

**Acknowledgements** I (lead author) would like to express my appreciation to Dr Giles Birchley, Dr Zuzana Deans, Dr Emanuele Valenti and Dr Iona Sherwood for the general advice, help and support they have given in conceiving this project.

**Contributors** SJO conceived the idea for the scoping review. All authors contributed to the development of the research questions and study design. SJO drafted the initial protocol with input from JI and RH. All authors contributed to redrafting and subsequent final revision of this protocol. SJO is the guarantor of the review.

**Funding** This work was supported by the Elizabeth Blackwell Institute and Wellcome Trust Institutional Strategic Support Fund, grant number 204813/Z/16/Z. The Elizabeth Blackwell Institute and Wellcome Trust were not involved in the development of the protocol other than providing financial support.

**Competing interests** None declared.

**Patient and public involvement** Patients and/or the public were not involved in the design, or conduct, or reporting, or dissemination plans of this research.

**Patient consent for publication** Not required.

**Provenance and peer review** Not commissioned; externally peer reviewed.

**ORCID iD**

Simon Jack Ogden http://orcid.org/0000-0002-6447-7903

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
