## [Reviewer comments · BMJ Open]

ARTICLE DETAILS

TITLE (PROVISIONAL)	Protocol for a scoping review to understand what is known about how GPs make decisions with, for and on behalf of patients who lack capacity
AUTHORS	Ogden, Simon Jack; Huxtable, Richard; Ives, Jonathan

VERSION 1 – REVIEW

REVIEWER	Dr Victoria Shepherd Cardiff University, UK
REVIEW RETURNED	13-Mar-2020

GENERAL COMMENTS	Thank you for an opportunity to review the protocol for a scoping review to understand what is known about how GPs make decisions with, for and on behalf of patients who lack capacity. The protocol describes a review on a topic that is important and of great relevance to clinicians and patients and their families alike, and those interested in the ethical and legal dimensions of best interests decision-making and mental capacity law. Overall the protocol is well written, the review well-designed, and the review is likely to make a useful contribution to the existing evidence on best interests decision-making in primary care. However, there are a few details which require minor revisions/clarifications. Legal jurisdiction: there is a curious focus on studies conducted only in England. Whilst limiting decisions made by GPs under the provisions of the MCA is clearly justified, it is not clear why only England is mentioned (e.g p5 line 6) despite it being Anglo-Welsh legislation, and why only studies conducted in England (and not those in Wales) are to be included. If studies in Wales are excluded, this should be justified in the protocol. Scope of decision-making: adults lacking capacity in primary care will have complex care and support needs as a result of their impairing condition, and many will require residential and nursing care. This means that many best interest decisions in primary care will often involve people living in care homes, or concern decisions made about the need to move into a residential facility, which is not acknowledged in the protocol. Importantly, this may impact on the relationship between the GP and patient. The Study Rationale refers to the continuity in the relationship between GP and patient (p6 line 38) – however care home residents will often change (or be required to change) GPs as the care home is usually not in the same catchment area. Decision-making context will be important, and highlighting the unique context for many of these patient groups will be useful for readers. Critical appraisal: whilst formal critical appraisal of individual sources
--

	of evidence is considered optional in scoping review methodology (although opinions vary) an acknowledgement of this would be helpful for readers less familiar with scoping (as opposed to systematic) reviews. Typographical error: p6 line 43 word missing from 'Given these are often important decisions that have a significant impact both the patient and their family...'
--	--

REVIEWER	Pamela B. Teaster Center for Gerontology, Virginia Tech USA
REVIEW RETURNED	02-Apr-2020

GENERAL COMMENTS	Hello, Thank you for inviting me to review this article. I have never reviewed a protocol before. Actually, the authors have done a good job laying out the protocol. I recommend publication after the author(s) address a very few concerns. Pbt Specific Comments:  1. Pp. 4-5, explain who determines if a patient lacks capacity and how the determination is made. 2. Pp. 4-5, are you reviewing mainly literature on the topic in the UK? In the US, this varies by state, and likely by region or healthcare provider and institution, as I am sure you are aware. Also, are you planning to look for best practices and if yes, how? 3. P.8, for inclusion criteria, do you mean peer-reviewed papers? If yes, what is the time frame that you would consider for these and other documents? 4. P.8, for your consideration, would you be reviewing relevant policies or regulations within some healthcare systems? Would these differ for type of healthcare delivery/ setting—community setting, hospital, hospice, long-term care facility? 5. What types of GP decisions will you include in your review—admission or discharge, medical procedure, end of life? 6. Would you consider in your review any white papers on the topic, should they exist? 7. P.8, who will be conducting the content analysis, and if there is more than one person involved, how will you resolve disagreements?
--

REVIEWER	Silvana Mengoni University of Hertfordshire, UK
REVIEW RETURNED	21-May-2020

GENERAL COMMENTS	This is a clearly written and informative protocol for a scoping review on an interesting topic. I look forward to reading the results of the review. I have suggested a few points of feedback for your consideration Abstract  1. P4 - I think it would be helpful to clearly state the research question (p7, line 29-30)
--

	Strengths and limitations 2. P4 lines 46-47 – does ‘While the study will focus on GPs in England’ refer to the later phases of the research programme as described in lines 35-37? My understanding is that the scoping review is not restricted to England, therefore this is confusing. I would suggest removing this part of the sentence Study Objectives 3. P6, line 53 – please clarify whether ‘knowledge gaps’ refers to GP’s knowledge gaps or knowledge gaps in the literature 4. P6, lines 48-50 – thus far, the focus has been on GPs but the study objective (along with the research question and inclusion criteria) focuses on ‘primary care’ where other healthcare professionals might be making decisions, not just GPs. Are these to be included also? If so, please can you incorporate this into the introduction to ‘set the scene’? Methods and analysis 5. P8, line 7 – please clarify what the ‘6 software’ is 6. P8, line 13 – approximately what proportion of the included and excluded papers will the second reviewer screen? 7. P8, line 29 – did your initial searches suggest that national guidelines are represented in your proposed data sources? 8. P9, line 6 – typo, ‘include’ should read ‘includes’ 9. P9, line 58 – typo, ‘GP’s’ should read ‘GPs’ Reporting checklist 10. P13, line 13-15 – if the funder, sponsor or institution did not have a role in developing the protocol, then I recommend stating this in the manuscript for transparency 11. P14, line 12-15 – please clarify where the ‘risk of bias in individual studies’ is discussed
--	--

VERSION 1 – AUTHOR RESPONSE

Reviewer(s)' Comments to Author:

Reviewer: 1

Reviewer Name: Dr Victoria Shepherd

Institution and Country: Cardiff University, UK

Please state any competing interests or state ‘None declared’: None declared

Please leave your comments for the authors below

Thank you for an opportunity to review the protocol for a scoping review to understand what is known about how GPs make decisions with, for and on behalf of patients who lack capacity. The protocol describes a review on a topic that is important and of great relevance to clinicians and patients and their families alike, and those interested in the ethical and legal dimensions of best interests decision-making and mental capacity law. Overall the protocol is well written, the review well-designed, and the review is likely to make a useful contribution to the existing evidence on best interests decision-making in primary care. However, there are a few details which require minor revisions/clarifications.

Legal jurisdiction: there is a curious focus on studies conducted only in England. Whilst limiting decisions made by GPs under the provisions of the MCA is clearly justified, it is not clear why only England is mentioned (e.g p5 line 6) despite it being Anglo-Welsh legislation, and why only studies conducted in England (and not those in Wales) are to be included. If studies in Wales are excluded, this should be justified in the protocol.

Thank you for highlighting this omission. It is clearly an important point that we have rectified in the paper. Indeed, we are planning on including GPs, allied primary care health care professionals, patients and their families who live in Wales in our future planned work. P1, line 4, 12, P2 line 36.

Scope of decision-making: adults lacking capacity in primary care will have complex care and support needs as a result of their impairing condition, and many will require residential and nursing care. This means that many best interest decisions in primary care will often involve people living in care homes, or concern decisions made about the need to move into a residential facility, which is not acknowledged in the protocol. Importantly, this may impact on the relationship between the GP and patient. The Study Rationale refers to the continuity in the relationship between GP and patient (p6 line 38) – however care home residents will often change (or be required to change) GPs as the care home is usually not in the same catchment area. Decision-making context will be important, and highlighting the unique context for many of these patient groups will be useful for readers.

This is an important point and there is much work on the decision to admit patients both to hospital and care homes. As per your suggestion, we have included some wording to this effect in the paper. P2 line 9

Critical appraisal: whilst formal critical appraisal of individual sources of evidence is considered optional in scoping review methodology (although opinions vary) an acknowledgement of this would be helpful for readers less familiar with scoping (as opposed to systematic) reviews.

We have acknowledged this in the paper now. P6 line 23

Typographical error: p6 line 43 word missing from 'Given these are often important decisions that have a significant impact both the patient and their family...'

Amended

Reviewer: 2

Reviewer Name: Pamela B. Teaster

Institution and Country: Center for Gerontology, Virginia Tech

USA

Please state any competing interests or state 'None declared': None declared

Please leave your comments for the authors below

Hello,

Thank you for inviting me to review this article. I have never reviewed a protocol before. Actually, the authors have done a good job laying out the protocol. I recommend publication after the author(s) address a very few concerns.

Pbt

Specific Comments:

1. Pp. 4-5, explain who determines if a patient lacks capacity and how the determination is made.

Thank you for highlighting this, we have included a small section on this. More broadly however, this work will look at how decisions are made once capacity has been assessed as lacking, rather than the process of assessing capacity itself. P2 line 13

2. Pp. 4-5, are you reviewing mainly literature on the topic in the UK? In the US, this varies by state, and likely by region or healthcare provider and institution, as I am sure you are aware. Also, are you planning to look for best practices and if yes, how?

This shall be an international search that aims to compare and contrast different jurisdictions, despite the scoping review being the foundations for a further qualitative study based in England and Wales. We are looking at the evidence we have for what is currently done, however, this is likely to include best practices as current practice often follows best practice. As such normative research will also be included. We can see there was some inconsistency in this for the paper and have amended it to make it clearer.

3. P.8, for inclusion criteria, do you mean peer-reviewed papers? If yes, what is the time frame that you would consider for these and other documents?

I have altered the wording to state peer-reviewed papers only P5 line 37. The final inclusion criteria states that papers from 1960 onwards will be included.

4. P.8, for your consideration, would you be reviewing relevant policies or regulations within some healthcare systems? Would these differ for type of healthcare delivery/ setting—community setting, hospital, hospice, long-term care facility?

We have kept the inclusion and exclusion criteria intentionally broad to capture as much information as possible which we would hope would cover all different healthcare settings in which primary care physicians and allied healthcare professionals work, and would expect this to capture some policies and regulations, most likely through the screening of refereneces.

5. What types of GP decisions will you include in your review—admission or discharge, medical procedure, end of life?

Our aim is to capture all of the above, we hope that p2 line 9 illustrates the variety of decisions being considered by the scoping review.

6. Would you consider in your review any white papers on the topic, should they exist?

By white paper do you mean a paper advocating a certain position is the best way to go rather than the UK government use of the term White paper which suggests law changes? We feel that given the ambiguity of the terms use internationally could add confusion and hope that our broad inclusion criteria might capture any literature including white papers of any description.

7. P.8, who will be conducting the content analysis, and if there is more than one person involved, how will you resolve disagreements?

The text has been amended to make this clearer p6 line 23

Reviewer: 3

Reviewer Name: Silvana Mengoni

Institution and Country: University of Hertfordshire, UK

Please state any competing interests or state 'None declared': None declared

Please leave your comments for the authors below

This is a clearly written and informative protocol for a scoping review on an interesting topic. I look

forward to reading the results of the review.

I have suggested a few points of feedback for your consideration

Abstract

1. P4 - I think it would be helpful to clearly state the research question (p7, line 29-30)

We agree, we have reworked the strengths and limitations section to include the research question.
P4

Strengths and limitations

2. P4 lines 46-47 – does ‘While the study will focus on GPs in England’ refer to the later phases of the research programme as described in lines 35-37? My understanding is that the scoping review is not restricted to England, therefore this is confusing. I would suggest removing this part of the sentence

We agree this may have made for some confusion so have amended it.

Study Objectives

3. P6, line 53 – please clarify whether ‘knowledge gaps’ refers to GP’s knowledge gaps or knowledge gaps in the literature

We have amended this by adding a statement to make it clear that we are looking at gaps in the literature rather than gaps in the knowledge of GPs or other healthcare professionals.

4. P6, lines 48-50 – thus far, the focus has been on GPs but the study objective (along with the research question and inclusion criteria) focuses on ‘primary care’ where other healthcare professionals might be making decisions, not just GPs. Are these to be included also? If so, please can you incorporate this into the introduction to ‘set the scene’?

We agree this and have amended the introduction to make this clearer. However, there will still be a GP-focussed slant, given that they make the majority of these decisions.

Methods and analysis

5. P8, line 7 – please clarify what the ‘6 software’ is

Amended – EndNote software, thank you for pointing this omission out.

6. P8, line 13 – approximately what proportion of the included and excluded papers will the second reviewer screen?

10% of papers returned in the search will be also be screen by a second reviewer. We have amended the paper to make this clearer

7. P8, line 29 – did your initial searches suggest that national guidelines are represented in your proposed data sources?

We found that while the database searches themselves did not themselves return national guidelines, when we searched the references of the included papers they were represented.

8. P9, line 6 – typo, ‘include’ should read ‘includes’

Amended, thank you

9. P9, line 58 – typo, 'GP's' should read 'GPs'

Amended

Reporting checklist

10. P13, line 13-15 – if the funder, sponsor or institution did not have a role in developing the protocol, then I recommend stating this in the manuscript for transparency

Amended

11. P14, line 12-15 – please clarify where the 'risk of bias in individual studies' is discussed

Given this is a scoping review and all papers were included without assessing for bias we did not include any section on individual paper bias.

VERSION 2 – REVIEW

REVIEWER	Victoria Shepherd Cardiff University, UK
REVIEW RETURNED	01-Aug-2020

GENERAL COMMENTS	The authors have responded carefully and appropriately to the reviewers' comments which has improved the manuscript. There are just a few minor areas that require revision to clarify where revisions have been made. I wish the authors every success with the scoping review and look forward to reading the findings with interest. In 'study selection' section it now states that 'all papers found to fit the inclusion criteria will be included in the review, which will reduce the risk of bias from the reviewers' (p.22 line 13). This is an odd statement to make and suggest removing it (I agree with the comments from the co-authors which are unfortunately still left in the manuscript!). It repeats the previous sentence that all papers meeting the criteria will be included and does not address the issue of risk of bias from individual papers (which I do not think is an issue but a more appropriate statement could be made about this). In 'study rationale' section it states ' We hope to find out what is understood about how decisions are or should be made in primary care with, for and on behalf of patients who lack capacity'. The rest of the paper is in the third person and so this could be amended accordingly. I notice a typographical error in the abstract introduction section 'healthc are'.
---

REVIEWER	Pamela B. Teaster Center for Gerontology, Virginia Tech, United States
REVIEW RETURNED	17-Aug-2020

GENERAL COMMENTS	The authors are proposing to conduct an important scoping review of the literature on making decisions on behalf of others, particularly concerning how GPs make decisions for people determined to lack capacity to do so for themselves. I only a few suggestions for
---

	improvement. Specific Comments:  1. Pp. 3-4, first example, line 14, there are a few places where you have a singular noun and a plural pronoun. Here is an example-- ...but a GP may become involved if they feel.... Change to ...if he or she feels. Same problem line 23-24--...if a patient lacks the capacity to make a particular decision, then it can be made for (should be him or her, not them) in his or her best interest. Same problem line 31. 2. Study rationale—are you examining how decisions are made for adult patients who lack capacity or children who are patients as well as adult patients? The way decisions are made vary somewhat. For example, for younger people, there is the ethical concept of fair innings that will come into play. 3. Protocol design—what is the time frame for your work? 4. Stage 1—identify research question—does primary care include long-term care or rehabilitation? One other issue you might consider in Stage 1—what is the length of time allotted for making decisions? 5. Stage 3—inclusion criteria, will you include commentary or opinion pieces on the topic of making decisions when patients lack capacity? 6. Stage 3—inclusion criteria, why are you choosing papers published since 1960? Consider truncating the time frame of your review, unless you are doing a historical one. Why not the last 10 or 20 years instead? 7. Ethics and dissemination—will you present at conferences in the UK only?
--	--

REVIEWER	Silvana Mengoni University of Hertfordshire, UK
REVIEW RETURNED	04-Aug-2020

GENERAL COMMENTS	Thank you for revising the manuscript and addressing my comments. I have one minor request for clarification. In the PRISMA-P reporting checklist, the authors have noted that the manuscript states that 'risk of bias in individual studies' was discussed on p7, and in my original review I requested that the authors clarify where this was discussed in the manuscript. In the reply to reviewers, the authors have stated that all papers were included without assessing for bias. If this is the case, then the PRISMA-P reporting checklist needs to be amended to reflect that this is not discussed in the protocol. However I also note that a statement has been added which seems to aim to address this (p22 of the combined proof, lines 11-13) – 'Given that this is a time -limited scoping review looking at the breadth of research rather than the quality of the research, all papers found to fit the inclusion criteria will be included in the review, which will reduce the risk of bias from the reviewers.' I am unclear as to the rationale for not assessing risk of bias of individual studies i.e. due to risk of bias from the reviewers. Why would the reviewers be biased? Tools exist to check quality of studies, reducing any potential bias from reviewers, but I am not suggesting that you need to do this. Although, scoping studies do not typically examine the quality of research to the same extent as systematic reviews or exclude studies based on their quality, they
--

may still consider this as part of a conclusion or synthesis.

VERSION 2 – AUTHOR RESPONSE

Reviewer 1

In 'study selection' section it now states that 'all papers found to fit the inclusion criteria will be included in the review, which will reduce the risk of bias from the reviewers' (p.22 line 13). This is an odd statement to make and suggest removing it (I agree with the comments from the co-authors which are unfortunately still left in the manuscript!). It repeats the previous sentence that all papers meeting the criteria will be included and does not address the issue of risk of bias from individual papers (which I do not think is an issue but a more appropriate statement could be made about this).

I apologise, we had meant to remove this on the previous submission but unfortunately trying to track changes on an iPad while on holiday got the better of me. I hope we have rectified this satisfactorily.

In 'study rationale' section it states ' We hope to find out what is understood about how decisions are or should be made in primary care with, for and on behalf of patients who lack capacity'. The rest of the paper is in the third person and so this could be amended accordingly.

We have amended accordingly, thank you.

I notice a typographical error in the abstract introduction section 'healthc are'.

Amended

Reviewer: 3

I have one minor request for clarification. In the PRISMA-P reporting checklist, the authors have noted that the manuscript states that 'risk of bias in individual studies' was discussed on p7, and in my original review I requested that the authors clarify where this was discussed in the manuscript. In the reply to reviewers, the authors have stated that all papers were included without assessing for bias. If this is the case, then the PRISMA-P reporting checklist needs to be amended to reflect that this is not discussed in the protocol.

Thank you for this, we have removed this from the PRISMA-P checklist.

However I also note that a statement has been added which seems to aim to address this (p22 of the combined proof, lines 11-13) – 'Given that this is a time -limited scoping review looking at the breadth of research rather than the quality of the research, all papers found to fit the inclusion criteria will be included in the review, which will reduce the risk of bias from the reviewers.'

As discussed above in response to Reviewer 1, we had meant to remove this part following discussion between authors, however it was accidentally not removed in the last submission. I hope now we have rectified this.

I am unclear as to the rationale for not assessing risk of bias of individual studies i.e. due to risk of bias from the reviewers. Why would the reviewers be biased? Tools exist to check quality of studies, reducing any potential bias from reviewers, but I am not suggesting that you need to do this. Although, scoping studies do not typically examine the quality of research to the same extent as systematic reviews or exclude studies based on their quality, they may still consider this as part of a conclusion or synthesis.

I agree and now understand that the comment suggesting reviewer and/or selection bias is incorrect and has been removed accordingly. We shall consider the quality of the studies when we perform the summarising and synthesising stage of the scoping review. Thank you for your guidance on this.

Reviewer: 2

1. Pp. 3-4, first example, line 14, there are a few places where you have a singular noun and a plural pronoun. Here is an example--...but a GP may become involved if they feel.... Change toif he or she feels. Same problem line 23-24--...if a patient lacks the capacity to make a particular decision, then it can be made for (should be him or her, not them) in his or her best interest. Same problem line 31.

Thank you for this comment, I suppose the use of the singular pronoun “they” is increasingly in use in both formal and informal writing and I think has been accepted by modern texts such as The Cambridge Guide to English Usage (2004, Cambridge University Press). Does it to some extent also conform to modern gender neutral rules? Nevertheless, I am sure you are technically correct and we are more than happy to change this. Thank you.

2. Study rationale—are you examining how decisions are made for adult patients who lack capacity or children who are patients as well as adult patients? The way decisions are made vary somewhat. For example, for younger people, there is the ethical concept of fair innings that will come into play.

We are looking at children and adults, and of course both groups present different challenges and are approached in different ways.

3. Protocol design—what is the time frame for your work?

This would be useful to include so have done so. We hope to have it completed within 6 months.

4. Stage 1—identify research question—does primary care include long-term care or rehabilitation? One other issue you might consider in Stage 1—what is the length of time allotted for making decisions?

Given the often chronic nature of the illnesses that are treated in primary care we are certain that our broad search will bring back results pertaining to long-term care. As alluded to in the ‘study rationale’ section whether a patient is admitted to hospital or a care home are also decisions that we will be considering.

5. Stage 3—inclusion criteria, will you include commentary or opinion pieces on the topic of making decisions when patients lack capacity?

Provided the commentary or opinion piece is published in a peer-reviewed journal it will be included. I have amended out inclusion and exclusion criteria to reflect this. Thank you for helping give this section more clarity.

6. Stage 3—inclusion criteria, why are you choosing papers published since 1960? Consider truncating the time frame of your review, unless you are doing a historical one. Why not the last 10 or 20 years instead?

We did discuss the time frame for paper inclusion, we decided that given theoretical arguments are just as valid were they published in 1960 as they are now we would keep the time frame wide. This will also allow us to include a larger volume of research in an area which is quite specific and has limited previous publications.

7. Ethics and dissemination—will you present at conferences in the UK only?

Given the recent COVID pandemic and resulting increase in virtual conferences we can now include this in the dissemination section.